# Immunogenicity and Safety of SARS-CoV-2 Protein Subunit Recombinant Vaccine (IndoVac^®^) as a Booster Dose against COVID-19 in Indonesian Adults

**DOI:** 10.3390/vaccines12050540

**Published:** 2024-05-14

**Authors:** Kusnandi Rusmil, Eddy Fadlyana, Rodman Tarigan Girsang, Riyadi Adrizain, Andri Reza Rahmadi, Hendarsyah Suryadinata, Muhammad Gilang Dwi Putra, Frizka Primadewi Fulendry, Dinda Tiaraningrum Nashsyah, Rona Kania Utami, Behesti Zahra Mardiah, I Gusti Ayu Trisna Windiani, I Gusti Agung Ngurah Sugitha Adnyana, Ni Luh Sukma Pratiwi Murti, I Ketut Agus Somia, I Made Susila Utama, Soetjiningsih Soetjiningsih, Ulfa Luthfiani Nurkamila Mutiara, Mita Puspita

**Affiliations:** 1Department of Child Health, Faculty of Medicine, Universitas Padjadjaran/Hasan Sadikin Hospital, Bandung 40161, Indonesia; kusnandi.rusmil2017@unpad.ac.id (K.R.); rodman.tarigan@unpad.ac.id (R.T.G.); riyadi2018@unpad.ac.id (R.A.); gilangdwiputra@gmail.com (M.G.D.P.); frizkaapf@gmail.com (F.P.F.); dindanashsyah@gmail.com (D.T.N.); ronakania@gmail.com (R.K.U.); behestizahramardiah@gmail.com (B.Z.M.); 2Department of Internal Medicine, Faculty of Medicine, Universitas Padjadjaran/Hasan Sadikin Hospital, Bandung 40161, Indonesia; rahmadiandri@yahoo.com (A.R.R.); hendarsyahsuryadinata@gmail.com (H.S.); 3Department of Child Health, Faculty of Medicine, Universitas Udayana, Prof. I.G.N.G Ngoerah Hospital, Denpasar 80114, Indonesia; trisnawindianidr@yahoo.co.id (I.G.A.T.W.); sugad168@yahoo.com (I.G.A.N.S.A.); sukma.pratiwi2509@gmail.com (N.L.S.P.M.); prof.soetji@gmail.com (S.S.); 4Department of Internal Medicine, Faculty of Medicine, Universitas Udayana, Prof. I.G.N.G Ngoerah Hospital, Denpasar 80114, Indonesia; agus_somia@yahoo.co.id (I.K.A.S.); susila_utama@unud.ac.id (I.M.S.U.); 5Global Clinical Development Division, PT Bio Farma, Bandung 40161, Indonesia; ulfa.luthfiani@biofarma.co.id (U.L.N.M.); mita.puspita@biofarma.co.id (M.P.)

**Keywords:** COVID-19, immunogenicity, protein recombinant vaccine (IndoVac^®^), safety

## Abstract

According to the WHO target product profile for COVID-19 vaccines, the vaccine in development should be indicated for active immunisation in all populations. Therefore, PT Bio Farma developed a candidate vaccine in a subunit protein recombinant platform to help overcome the issue. This trial was an observer-blind, randomised, prospective intervention study. This study targeted individuals who had received complete primary doses of the authorised/approved COVID-19 vaccine. The groups were divided into the primary inactivated vaccine (CoronaVac^®^) group, the primary viral vector vaccine (ChAdOx1) group, and the primary mRNA vaccine (BNT162b2) group that received the recombinant protein (IndoVac^®^). The groups were compared with the control and primary mRNA vaccine (BNT162b2). The participants enrolled in the study were from two primary care centres in Bandung City and three primary care centres in Denpasar City. A total of 696 participants were enrolled from 1 September to 31 October 2022. The demographic characteristics of the all-vaccine group showed a uniform distribution. The results showed that, compared with the control, the investigational product had inferior effectiveness 14 days after the booster dose was administered. However, 28 days after the booster dose, the investigational product exhibited non-inferior effectiveness compared with the primary groups that received CoronaVac^®^ (GMR 0.76 (0.57–0.99)) and ChAdOx1 (GMR 0.72 (0.56–59.93)), but the BNT162b2 group (GMR 0.61 (0.39–0.94)) was inferior to the control. At 12 months follow-up after the booster dose, three serious adverse events (SAEs) were reported in three participants, with causality not correlated with the investigated products. Neither AEs of special interest nor severe COVID-19 cases were reported throughout the follow-up period; thus, the IndoVac^®^ vaccine as a booster was immunogenic and safe. Until the 6-month follow-up after the booster dose, the IndoVac^®^ vaccine was well tolerated and all reported AEs resolved. This vaccine is registered and can be included in the immunisation programme.

## 1. Introduction

According to the WHO target product profile for COVID-19 vaccines, the vaccine in development should be indicated for active immunisation in all populations, including the elderly, in areas with an ongoing COVID-19 outbreak, and should be used in conjunction with other control measures to curtail or end such an outbreak [1,2,3,4]. The investment in SARS-CoV-2 vaccine development will contribute to the stabilisation of the supply of safe and affordable COVID-19 vaccines in the global market, particularly for distribution in low- and middle-income countries [2,4,5,6]. The COVID-19 pandemic and the precipitously increased number of deaths worldwide necessitate the urgent development of SARS-CoV-2 vaccines, hence requiring a new pandemic paradigm [1,3,4,5].

In Indonesia, 30 clinical trials for COVID-19 vaccines were conducted, and 13 vaccines were subsequently approved for use [5,6,7,8,9]. The national vaccination campaign for COVID-19 was introduced in January 2021, prioritising healthcare workers in the first stage and continuing to essential public service workers and older people [5,6,10]. Indonesia targeted to vaccinate at least 80% of its population by the end of 2023. Regarding the COVID-19 vaccination status in Indonesia, >80% of the population are vaccinated with the first dose and >70% with the second dose; however, <40% have received the first booster and only 2% have received the second booster [6,7,8,9,10].

The Ministry of Health in Indonesia was involved in the development and implementation of the COVID-19 vaccination programme, including the IndoVac^®^ vaccine [11]. IndoVac^®^ is a domestically produced COVID-19 vaccine developed by PT Bio Farma, a state-owned enterprise in Indonesia [11,12]. PT Bio Farma developed a candidate vaccine using a subunit protein recombinant platform to help overcome the issue. IndoVac^®^ has recently been authorised for use in Indonesia following clinical trials [11,12,13]. In a phase 3 clinical trial comparing the primary doses of IndoVac^®^ with Covovax, 2070 participants were included in the immunogenicity group [11,12,13]. IndoVac^®^ demonstrated non-inferior immunogenicity to Covovax when assessed for neutralising antibody titres against the delta variant. The results of the clinical trial of IndoVac^®^ as a booster are now available [11,12,13]. As a locally produced vaccine, IndoVac^®^ has clear benefits for use in Indonesia [11,12,13].

Preclinical animal studies and controlled trials have demonstrated the potential utility of subunit protein vaccines, making them one of the most viable options for resource-limited settings because they can be stored in refrigerators and distributed easily [14,15]. This study aimed to evaluate the non-inferiority of the immune response of the SARS-CoV-2 neutralising antibody in IndoVac^®^ to the control vaccine 14 days after administration of the booster dose.

## 2. Materials and Methods

### 2.1. Study Design

This multicentre, observer-blind, randomised, prospective intervention study was planned for 900 individuals who had received complete primary doses of the authorised/approved COVID-19 vaccine. They willingly participated in the booster study and signed consent forms. Participants were divided into six groups, each with 150 participants per arm who had received a complete primary dose of an inactivated vaccine (CoronaVac^®^), mRNA vaccine (BNT162b2), or viral vector vaccine (ChAdOx1). Furthermore, they received one booster dose of the SARS-CoV-2 subunit protein recombinant vaccine (IndoVac^®^) or the active control mRNA vaccine (BNT162b2). This study is registered at clinicaltrial.gov, ID number NCT05525208.

### 2.2. Study Participants

A total of 900 participants were enrolled in the study and divided into the primary inactivated vaccine (CoronaVac^®^) group, the primary viral vector vaccine (ChAdOx1) group, and the mRNA vaccine (BNT162b2) group, with 300 participants each. After being informed about the study, signed informed consent or signed assent and informed consent was obtained from the participants. The primary inclusion criteria were adults aged ≥18 years and compliance with study instructions and trial schedules. The exclusion criteria were as follows: presenting with mild, moderate, or severe illness with a fever (axillary temperature ≥ 37.5 °C), history of uncontrolled asthma, history of allergy to vaccines or vaccine ingredients, and severe adverse reactions to vaccines, such as urticaria, dyspnoea, and angioneurotic oedema. The participants were enrolled from two primary healthcare (PHC) centres in Bandung City (Ibrahim Adjie PHC and Dago PHC) and three PHCs in Denpasar City (Denpasar Selatan PHC and two Denpasar Utara PHC) between September and October 2022.

### 2.3. Randomisation and Blinding

The participants were randomised into each treatment group. The investigator strictly followed the randomisation list provided by PT Bio Farma. Treatment was allocated based on the randomisation list so that each randomisation number corresponded to only one strictly randomly assigned treatment. A total of 900 participants were randomised, with the following inclusion numbers and allocated vaccines: 001–300 (CoronaVac^®^), 301–600 (ChAdOx1), and 601–900 (BNT162b2), and randomisation codes (A or B).

### 2.4. Vaccines

The investigational product was IndoVac^®^ (SARS-CoV-2 subunit protein recombinant vaccine). Each 0.5 mL dose of the vaccine contains 25 μg of SARS-CoV-2 RBD subunit recombinant protein, 750 μg of aluminium as an adjuvant, 750 μg of CpG 1018 as an adjuvant, 2.226 mg of NaCl, and 0.923 mg of tris(hydroxymethyl)aminomethane (Batch no. 24800222 and 24800422). The comparative product was the Pfizer-BioNTech^®^ COVID-19 vaccine. One dose (0.3 mL) contains 30 μg of COVID-19 mRNA vaccine (embedded in lipid nanoparticles).

Single-stranded 5′-capped messenger RNA (mRNA) was produced using cell-free in vitro transcription from the corresponding DNA templates encoding the viral spike (S) protein of SARS-CoV-2. Sterile concentrate was used for dispersion. The Batch no. were FT5335 and P0001431.

### 2.5. Sample Size and Study Analysis

The sample size was calculated based on noninferiority tests for the ratio of two means (log-normal data) using PASS. Based on WHO Guidelines on ‘Considerations for Evaluation of COVID-19 Vaccines’ (30 March 2022 version) [13], noninferiority to the comparator (EUL listed COVID-19 vaccine) was defined as the lower bound (LB) of 95% confidence interval (CI) in geometric mean titre (GMT) ratio (GMT in the new vaccine/GMT in the comparator vaccine) of >0.67. If the LB of the 95% CI in the GMT ratio of the vaccine candidate to the active control (R0) is 0.675, the actual ratio (R1) is 0.9, the coefficient of variation (COV) is 0.8, and the power is 90%, thus the minimum sample size required is 127 subjects per arm. With a vaccine-to-control ratio of 1:1 and a dropout rate of 10%, each arm required 140 participants. The sample size was also calculated using the assumption that the vaccine arm will show at least 88% seropositivity rate and assuming a maximum seropositivity rate among active controls of 98%, the study needs 135 and 135 participants in the SARS-CoV-2 vaccine and active control groups, respectively, to be able to reject the null hypothesis that the seropositivity rates for the experimental and control participants are equal with probability (power) of 0.9 and one-sided alpha of 0.025. Assuming a dropout rate of 10%, each arm would require 149 participants. To accommodate the minimum sample size required for the aforementioned evaluations, the study will require approximately 150 participants per arm. This phase II study had six treatment arms; therefore, the total sample size planned was 900 subjects. To evaluate this null hypothesis, uncorrected chi-squared statistics were used for each primary series vaccine group.

### 2.6. Immunogenicity Measurements

For primary evaluation criteria, antibody titres in subjects were measured at baseline (pre-vaccination) and 14 days after booster vaccination. For secondary evaluation criteria, additional blood samples were collected 28 days, 3 months, and 6 months after booster vaccination. IgG antibody titres were evaluated using CMIA (Chemiluminescent Microparticle Immunoassay) by the Prodia Laboratory located in each centre. The neutralizing antibody titres were evaluated using the surrogate Virus Neutralization Test (sVNT) and microneutralization assay by the clinical trial laboratory at Bio Farma. Seroconversion was evaluated following immunization with the SARS-CoV-2 vaccine and an active control. The neutralization assay and sVNT were conducted against a SARS-CoV-2 variant of concern (Omicron strain).

### 2.7. Safety Measurements

For safety measurements, the investigator assessed the intensity (code 1, 2, or 3), duration, and relation of each adverse event to the trial vaccines. Local and systemic reactions, expected or not, occurring within 30 min and 28 days after each immunization were evaluated by interviewing the subjects during post-surveillance visits. Serious adverse events were evaluated during the study, until 6 months after booster vaccination. Particularly, the body temperature was measured (using a thermometer) for seven days after vaccination, in the evening, and/or at the peak febrile time and the highest temperature was recorded in the diary card in degrees Celsius. The trial team then recorded the information in the electronic CRF.

## 3. Results

In this study, 743 participants were screened, of which 47 were excluded because of pregnancies, comorbidities, acute illness, plan to change domicile, and primary vaccination < 6 months. A total of 696 participants were enrolled in the study from 1 September to 31 October 2022 for the primary inactivated vaccine (CoronaVac^®^) group, the primary viral vector vaccine (ChAdOx1) group, and the primary mRNA vaccine (BNT162b2) group.

### 3.1. Demographics and Baseline Characteristics

The CoronaVac^®^ primary vaccine group was composed of 300 participants, including 112 male (37.3%) and 188 female (62.7%) participants, with a mean age of 37.93 ± 13.46 years. Most of the participants had finished senior high school (55.3%) and (68.3%) were employed. The participants came from various ethnic groups, mostly Sundanese (48.3%), Balinese (26.3%), and Javanese (14.3%).

The ChAdOx1 primary vaccine group had 300 participants, of which 139 were male (46.3%) and 161 were female (53.7%), with a mean age of 37.21 ± 12.33 years. Most of the participants had completed senior high school (53.3%) and were employed (73.3%). The participants came from various ethnic groups, mostly Balinese (46.7%), followed by Sundanese (31.7%) and Javanese (12.7%).

A total of 96 participants were enrolled in the BNT162b2 primary vaccine group: 37 male (38%) and 59 female (62%) participants with a mean age of 36.42 ± 11.56 years. The majority of the participants were senior high school graduates (52%) and employed (65%). The participants came from various ethnic groups, mostly Sundanese (73%) and Javanese (21%). The demographic characteristics of participants in all vaccine groups showed a fair distribution with regard to age, BMI, sex, previous education, employment, and ethnicity (Appendix A).

### 3.2. Immunogenicity

Immunogenicity was analysed in each group, with 95% CI provided, and the proportion of participants with seropositivity and seroconversion at each time point between the IndoVac^®^ vaccine and the control groups was compared using the chi-square test. Antibody persistence was evaluated by comparing neutralising and IgG antibody titres against the omicron strain 3 and 6 months after the booster dose in all primary vaccine groups (CoronaVac^®^, ChAdOx1, and BNT162b2) (Figure 1).

Overall, the results of this study showed that the IndoVac^®^ vaccine was inferior to the control 14 days after the booster dose. Moreover, 28 days after the booster dose, the IndoVac^®^ vaccine exhibited non-inferior effectiveness compared with the primary groups of CoronaVac^®^ (GMR 0.76 (0.57–0.99)) and ChAdOx1 group (GMR 0.72 (0.56–59.93)), but the BNT162b2 group (GMR 0.61 (0.39–0.94)) was inferior to the control.

The seropositivity rate in all primary vaccine groups (CoronaVac^®^, ChAdOx1, and BNT162b2) was still maintained compared with 14 or 28 days after the booster dose. The GMT was still above the cutoff value for seropositivity of the neutralising antibody (≥46.03) IU/mL and IgG antibody (≥7.1 BAU/mL). Regarding antibody persistence, no significant difference was observed for seropositive rates and GMT between the investigational product and control groups at 14 days, 28 days, 3 months, and 6 months.

A.CoronaVac^®^ primary vaccine group

In the CoronaVac^®^ primary vaccine group, the neutralising antibody GMTs (IU/mL) in the IndoVac^®^ vaccine group at baseline and 14 days, 28 days, 3 months, and 6 months after the booster dose were 147.52, 1266.69, 1455.61, 717.36, and 758.54, respectively. In the control group, the neutralising antibody GMTs (IU/mL) at baseline and 14 days, 28 days, 3 months, and 6 months after the booster dose were 151.75, 1982.51, 1926.17, 665.46, and 589.70, respectively. The seropositive rates at baseline and 14 days, 28 days, 3 months, and 6 months after the booster dose in the IndoVac^®^ vaccine group were 76.87%, 98.64%, 98.63%, 99.31%, and 99.29%, respectively. In the control group, the seropositive rates were 82.31%, 100%, 100%, 98.64%, and 99.31%, respectively (Figure 2).

In the CoronaVac^®^ primary vaccine group, the IgG antibody GMTs (BAU/mL) in the vaccine group at baseline and 14 days, 28 days, 3 months, and 6 months after the booster dose were 266.20, 2817.08, 2663.47, 1804.82, and 1090.63, respectively. In the control group, the neutralising antibody GMTs (BAU/mL) at baseline and 14 days, 28 days, 3 months, and 6 months after the booster dose were 266.39, 3482.59, 2546.08, 1135.17, and 629.27, respectively. The seropositive rates at baseline and 14 days, 28 days, 3 months, and 6 months after the booster dose in the IndoVac^®^ vaccine group were 98.64%, 100.00%, 100.00%, 100.00%, and 100.00%, respectively. In the control group, the seropositive rates were 99.32%, 100.00%, 100.00%, 100.00%, and 100%, respectively (Figure 2).

B.ChAdOx1 primary vaccine group

In the ChAdOx1 primary vaccine group, the neutralising antibody GMTs (IU/mL) at baseline and 14 days, 28 days, 3 months, and 6 months after the booster dose were 251.46, 1320.43, 1039.13, 1352.34, and 533.37, respectively. In the control group, the neutralising antibody GMTs (IU/mL) at baseline and 14 days, 28 days, 3 months, and 6 months after the booster dose were 250.80, 2142.50, 1445.64, 1115.78, and 503.63, respectively. In the IndoVac^®^ vaccine group, the seropositive rates at baseline and 14 days, 28 days, 3 months, and 6 months after the booster dose were 86.58%, 99.33%, 98.66%, 98.63%, and 98.62%, respectively. In the control group, the seropositive rates were 85.14%, 100.00%, 100.00%, 100.00%, and 99.31%, respectively (Figure 2).

In the ChAdOx1 primary vaccine group, the IgG antibody GMTs (BAU/mL) in the vaccine group at baseline and 14 days, 28 days, 3 months, and 6 months after the booster dose were 409.33, 2318.75, 2137.70, 1640.16, and 953.66, respectively. In the control group, the neutralising antibody GMTs (BAU/mL) at baseline and 14 days, 28 days, 3 months, and 6 months after the booster dose were 396.45, 3122.96, 2328.21, 1270.84, and 765.02, respectively. In the IndoVac^®^ vaccine group, the seropositive rates at baseline and 14 days, 28 days, 3 months, and 6 months after the booster dose were 100.0%, 100.0%, 100.00%, 100.00%, and 100.00% respectively. In the control group, the seropositive rates were 100.0%, 100.00%, 100.00%, 100.00%, and 100.00%, respectively (Figure 2).

C.BNT162b2 primary vaccine group

In the BNT162b2 primary vaccine group, the neutralising antibody GMTs (IU/mL) at baseline and 14 days, 28 days, 3 months, and 6 months after the booster dose were 237.49, 1012.50, 1019.69, 956.81, and 463.94, respectively. In the control group, the neutralising antibody GMTs (IU/mL) at baseline and 14 days, 28 days, 3 months, and 6 months after the booster dose were 254.56, 2332.31, 1661.66, 1033.56, and 416.53, respectively. In the IndoVac^®^ vaccine group, the seropositive rates at baseline and 14 days, 28 days, 3 months, and 6 months after the booster dose were 89.80%, 100.00%, 100.00%, 100.00%, and 100.00%, respectively; in the control group, the rates were 86.96%, 100.00%, 100.00%, 97.78%, and 95.56%, respectively (Figure 2).

In the BNT162b2 primary vaccine group, the IgG antibody GMTs (BAU/mL) in the IndoVac^®^ vaccine group at baseline and 14 days, 28 days, 3 months, and 6 months after the booster dose were 433.70, 2396.66, 2476.90, 1809.55, and 1012.94, respectively. In the control group, the neutralising antibody GMTs (BAU/mL) at baseline and 14 days, 28 days, 3 months, and 6 months after the booster dose were 500.11, 4531.76, 3699.89, 1937.69, and 973.59, respectively. The seropositive rates at baseline and 14 days, 28 days, 3 months, and 6 months after the booster dose in the vaccine group were 100.00%, 100.00%, 100.00%, 100.00%, and 100.00%, respectively; in the control group, the rates were 100.00%, 100.00%, 100.00%, 100.00%, and 100.00%, respectively (Figure 2).

### 3.3. Safety

#### 3.3.1. Adverse Events (AEs)

A.AEs in the CoronaVac^®^ primary vaccine group

The most reported solicited AE until 28 days after the booster dose in the CoronaVac^®^ primary vaccine group was local pain (vaccine group, 34%; control group, 40%), followed by myalgia (IndoVac^®^ vaccine group, 28%; control group, 39%). The incidence rates of unsolicited AEs were <6% in the vaccine group. Most of the AEs in all the groups were mild (Figure 3).

B.Aes in the ChAdOx1 primary vaccine group

The most reported solicited AE until 28 days after the booster dose in the ChAdOx1 primary vaccine group was local pain (vaccine group, 25%; control group, 35%), followed by myalgia (both groups, 23%). The incidence rates of unsolicited AEs were <5% in the IndoVac^®^ vaccine group. Most of the AEs in all the groups were mild (Figure 4).

C.AEs in the BNT162b2 primary vaccine group

The most reported solicited AE until 28 days after the booster dose in the BNT162b2 primary vaccine group was local pain (vaccine group, 45%; control group, 47%), followed by myalgia (IndoVac^®^ vaccine group, 33%; control group, 43%). The incidence rates of unsolicited AEs were <5% in the IndoVac^®^ vaccine group. Most of the AEs in all the groups were mild (Figure 5).

#### 3.3.2. Serious AEs (SAEs)

In total, three SAEs occurred in the study; however, they were not related to the investigational products. No AEs of special interest were observed during the follow-up period. No deaths were reported. SAEs were evaluated by the Data Management Safety Board.

## 4. Discussion

Indonesian people’s perspectives of booster programmes show the dominance of positive sentiments towards vaccine boosters compared with negative sentiments [5,16]. A study found a significant difference in vaccination coverage and vaccination timing between cities and rural areas [5,16]. Because of national mandatory regulations and the demand for COVID-19 booster vaccinations in Indonesia, PT Bio Farma has developed a COVID-19 vaccine to help accomplish target vaccination for all populations globally, particularly in Indonesia [11,12].

This vaccine candidate is based on SARS-CoV-2 recombinant protein-based RBD protein developed by Texas Children’s Hospital Center for Vaccine Development at Baylor College of Medicine. It is based on the sequence of the wild-type SARS-CoV-2 RBD amino acid, representing residues 331–549 of the spike (S) protein (GenBank: QHD43416.1) of the Wuhan-Hu-1 isolate (GenBank: MN908947.3) [17,18,19].

In this vaccine design, RBD was formulated with CpG 1018 and aluminium hydroxide adjuvants to effectively induce antibodies that neutralise wild-type live viruses while minimising Th2-biased responses with no vaccine-related AEs [14,15,17,18,19]. Recombinant protein vaccine containing SARS-CoV-2 RBD elicited a potent immune response, with neutralising antibodies reaching 99% seroconversion, although the response was lower in the group aged ≥65 years compared with the younger group. This vaccine also induced persistent antibody responses, lasting >6 months [20,21,22]. As RDB contains multiple conformational and conserved neutralising epitopes, the vaccine was predicted to have cross-neutralisation potency against virus variants [20,21,22].

The results of this study showed that the investigational products were inferior 14 days after the booster dose compared with the control. At 28 days after the booster dose, the investigational product was non-inferior in the primary group of CoronaVac^®^ (GMR 0.76 (0.57–0.99)) and the ChAdOx1 group (GMR 0.72 (0.56–59.93)) compared to the control. Based on the seropositive and seroconversion rates 28 days after the booster dose, no statistical difference was found between the IndoVac^®^ vaccine and the control vaccine despite the difference in GMT. A heterologous booster study was also conducted for Biological E’s Corbevax, another vaccine with a similar platform to IndoVac^®^ Vaccine, which has been approved by the Indian Regulatory Authority as a heterologous booster for Covishield (ChAdOx1 vaccine) and Covaxin (inactivated vaccine) [21]. A study in Iran showed a different result from our study [15]. The seropositive rate was only 30% after 28 days of the booster dose administration [15]. Other studies have also shown that antibody levels after administration of a booster dose of the mRNA vaccine were higher than after administration of the protein subunit recombinant and viral vector vaccine [22,23,24]. A study in the UK evaluating different vaccine platforms as third-dose boosters also demonstrated the potential of all the vaccines tested to boost immunity post-booster, as measured by anti-spike IgG and neutralising assays [25]. All vaccines were shown to boost immunity in older and younger populations [21,22,23,24,25]. However, marked differences were found in response to specific booster vaccines [21,22,23,24,25].

The GMTs at baseline in the IndoVac^®^ vaccine group were lower than those in the control group for all primary vaccine groups, particularly in the CoronaVac^®^ primary vaccine group. At 14 days after the booster dose, the GMTs in all groups increased; those in the control group were significantly higher than those in the IndoVac^®^ vaccine group (*p* < 0.05). However, the GMTs in the control group were still significantly higher than those in the vaccine group. Previous studies have shown that individuals who received the homologous inactivated virus vaccine as a primary vaccine exhibited lower binding antibodies than the other groups at baseline [20,21,24]. This result is consistent with the findings of a previous study suggesting a lower decrease in neutralising antibody titres after administration of the subunit protein vaccine booster compared with mRNA vaccines [20,26]. In addition, a study showed that the neutralising antibody (nAb) titre against SARS-CoV-2 variants after booster with mRNA vaccine was lower than the titres with the inactivated virus (CoronaVac^®^) and the protein recombinant vaccine (ChAdOx1) as the primary vaccine [20,21,24]. However, 28 days after the booster dose, all GMTs decreased gradually, and IgG levels were slightly reduced at 28 days [20,21,24].

At 3 and 6 months, the investigational product showed higher GMTs than the control, showing a significant difference. The significant difference in GMTs for recombinant COVID-19 vaccines at 3 and 6 months highlights the effectiveness of these vaccines in boosting immune responses over time. Studies have shown that administering a recombinant protein subunit vaccine booster, such as ZF2001, at 3- to 9-month intervals after the initial vaccination with an inactivated vaccine such as CoronaVac^®^ can lead to robust immune responses [27]. However, the NVX-CoV2373 vaccine has shown promising results in adolescents; this vaccine is immunogenic and has significant GMTs [28]. The GMTs of the recombinant COVID-19 vaccines at 3 and 6 months demonstrated sustained immunogenicity and efficacy of these vaccines, highlighting their role in boosting immune responses over time and providing protection against severe COVID-19 outcomes and emerging variants [27,28,29].

The development of recombinant COVID-19 vaccines and booster shots has shown promising results in terms of immunogenicity, efficacy against different variants, and durability of immune responses over time. These findings support the ongoing efforts to combat the COVID-19 pandemic through vaccination strategies that enhance immunity and provide long-term protection [30,31].

CoronaVac^®^ and BNT162b2 as primary vaccines and the IndoVac^®^ vaccine as the booster exhibited an approximate increase in GMTs. A previous study in Indonesia revealed results similar to those of this study: a significant increase in binding and neutralising antibodies after boosting with protein subunit recombinant vaccine such as ChAdOx1-S (ChAdOx1 heterologous booster), which means that the GMT obtained after a booster dose of mRNA vaccine was higher than that obtained after a booster dose of protein subunit recombinant and viral vector vaccine [26].

Antibody persistence 3 and 6 months after the booster dose was evaluated by analysing neutralising antibodies against the omicron variant and IgG antibody titres in all primary vaccine groups [32,33]. The neutralising antibody and IgG antibody titres in all primary vaccine groups were still maintained compared with the values 14 and 28 days after the booster dose. No significant differences were observed for seropositive rates and GMT between the IndoVac^®^ vaccine and control groups. Surprisingly, significant differences in the seropositive rates and GMT between investigational product and control groups were found in the CoronaVac^®^ and ChAdOx1 primary vaccine groups at 3 and 6 months.

The significant difference in IgG results for recombinant COVID-19 vaccines at 3 and 6 months indicates sustained humoral immune responses over time [27,34]. Studies have shown that recombinant protein subunit vaccines, such as ZF2001, can elicit robust IgG antibody responses against SARS-CoV-2, with significant efficacy in preventing COVID-19 and related severe outcomes [27]. In addition, the NVX-CoV2373 vaccine, a recombinant S protein vaccine co-formulated with a saponin-based adjuvant, demonstrated higher IgG levels [28]. These findings highlight the importance of recombinant protein subunit vaccines in boosting humoral immunity against COVID-19 and providing long-lasting protection against the virus and its variants [27,28,34,35].

This study revealed that the seropositive rates against the omicron variant and IgG antibody titres in all primary vaccine groups were still maintained compared with those at 14 and 28 days after the booster dose. The results showed that the immune response, which was evaluated by analysing neutralising antibodies against the omicron variant in all primary vaccine groups, gradually declined 12 months after the booster dose; however, the rates were not significant in both booster groups [36,37,38,39].

Several studies have investigated the persistence of antibodies after a booster vaccine for COVID-19 at 3, 6, and 12 months. In one study, the safety, immunogenicity, and antibody persistence of a bivalent beta-containing booster vaccine against COVID-19 were evaluated, revealing that the antibody response persisted for at least 3 months after the booster dose [40]. In another study, the duration of detectable anti-spike antibodies after COVID-19 vaccination was measured, indicating that participants maintained detectable anti-spike antibodies until dropout or censoring, reflecting a highly durable antibody response to COVID-19 vaccination [41]. A third study assessed the 3-month antibody persistence of a bivalent omicron-containing booster vaccine, revealing that the omicron-BA.1-containing bivalent booster consistently induced higher neutralising antibody titres against omicron BA.1, with no decrement in the response against ancestral SARS-CoV-2 (D614G) [42]. Mechanistic modelling projections were utilised in a fourth study to estimate antibody persistence after a single-dose vaccine, revealing that the projected overall 24-month persistence after a single dose was 70.5% for binding antibodies and 55.2% for neutralising antibodies [43]. These studies have provided insights into antibody persistence following COVID-19 booster vaccination at 12 months, suggesting that homologous booster regimens can lead to a significant increase in antibody persistence [40,41,42,43].

This study showed that the microneutralisation and IgG levels of the investigational product were not different 28 days, 3 months, and 6 months after the booster dose compared with the control. A significant difference was observed for the seropositive rates and GMTs between the investigational product and the control groups at 3 and 6 months with the primary vaccines CoronaVac^®^ and ChAdOx1. A microneutralisation assay for the IndoVac^®^ vaccine showed a significant increment in immunogenicity against the omicron variant compared to the prebooster baseline [36,37,38,39].

Local pain, followed by myalgia, was the most prevalent AE reported in both vaccine and control groups of all primary vaccine groups. This finding was consistent with those of several studies reporting that the most common side effects were pain at the injection site (for local AEs) and myalgia (for the most common systemic AEs) [20,21,22,23,24,26,44]. Three SAEs were reported, with causality not related to the study products.

Thus, based on this study, IndoVac^®^ was safe during the safety follow-up 28 days after the booster dose and showed mild-to-moderate, transient reactogenicity. No safety concerns related to vaccination were seen, as indicated by the similar percentages of participants reporting unsolicited AEs in the vaccine and control groups. IndoVac^®^ (subunit protein recombinant vaccine platform) can be a good candidate for booster doses of primary CoronaVac^®^, ChAdOx1, and BNT162b2 vaccines [11,12]. Considering similar results from other studies, these data provide immunisation advisory committees and policymakers with additional immunological and reactogenicity information, which could allow flexibility in deploying heterologous booster vaccines. Other considerations should include clinical, logistical, and supply considerations targeted to the populations with the greatest need.

Limitations: This study used the BNT162b2 vaccine as a control product, which uses a different platform from IndoVac^®^. The available COVID-19 booster vaccines that had been authorised for emergency use in Indonesia at the time of the study were BNT162b2 and ChAdOx1. ChAdOx1 was unavailable, and other subunit protein recombinant vaccines (e.g., Covovax) had not received EUA as a booster dose in Indonesia. The minimum sample size in the BNT162b2 primary vaccine group was not met because most of the population in the study area did not receive the BNT162b2 vaccine as their primary COVID-19 vaccine.

## 5. Conclusions

IndoVac^®^ has favourable immunogenicity and safety profile as a booster dose in participants who previously received primary doses of CoronaVac^®^, ChAdOx1, or BNT162b2. IndoVac^®^ was inferior to the control 14 days after the booster but was non-inferior after the booster with primary CoronaVac^®^ and ChAdOx1 vaccines. The vaccine has favourable immunogenicity based on seropositive and seroconversion rates 14 and 28 days after the booster dose. IndoVac^®^ showed a significant increment in immunogenicity against the omicron variant at 3 and 6 months based on the prebooster baseline. The IndoVac^®^ vaccine was well tolerated until the 6-month follow-up after the booster dose, and all reported AEs resolved.

## Figures and Tables

**Figure 1 vaccines-12-00540-f001:**
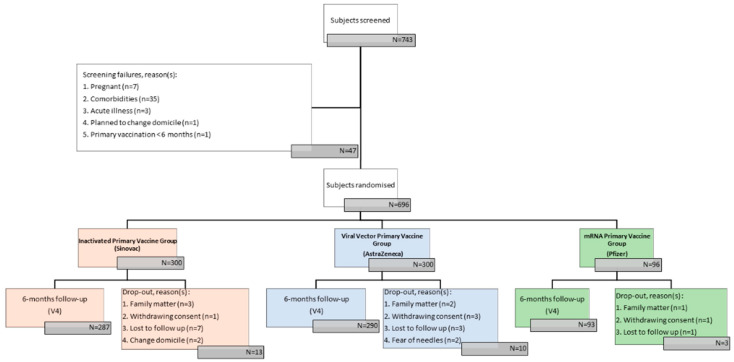
Participant selction.

**Figure 2 vaccines-12-00540-f002:**
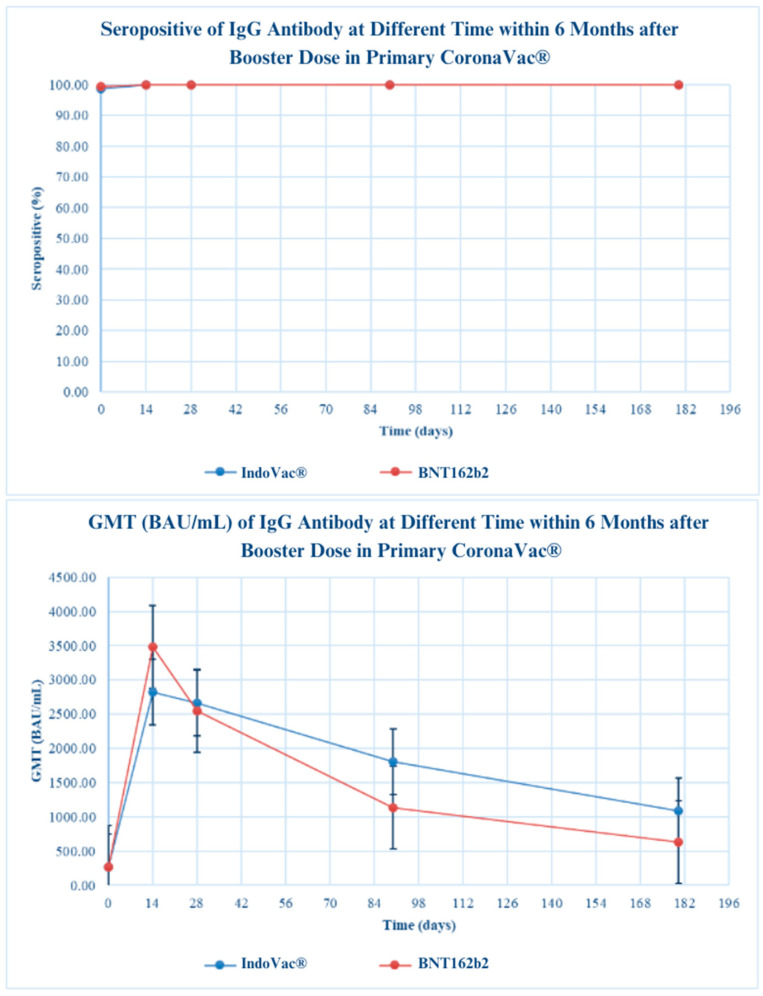
Evaluation of immunogenicity over 6 months after administration of booster dose.

**Figure 3 vaccines-12-00540-f003:**
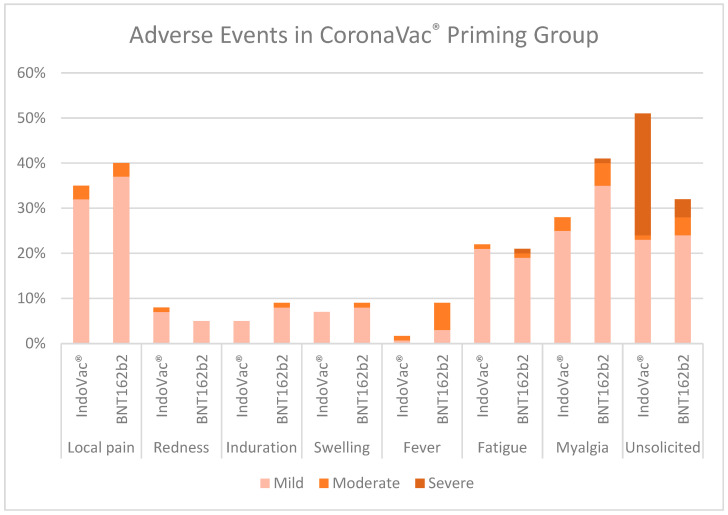
Adverse events in the CoronaVac^®^ primary vaccine group.

**Figure 4 vaccines-12-00540-f004:**
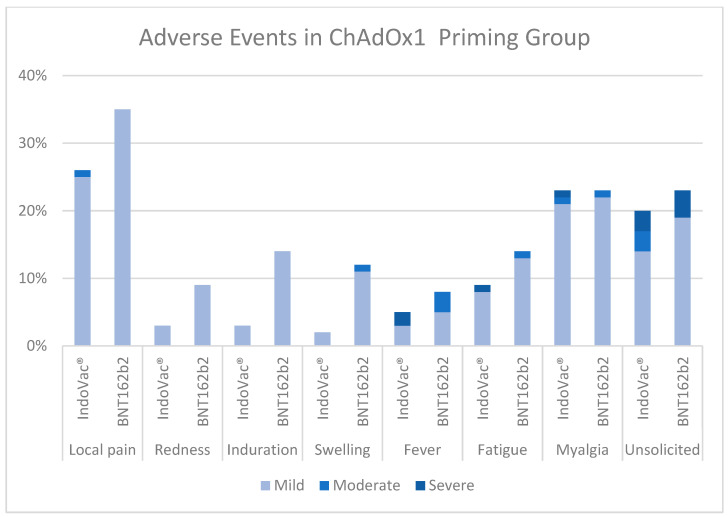
Adverse events in the ChAdOx1 primary vaccine group.

**Figure 5 vaccines-12-00540-f005:**
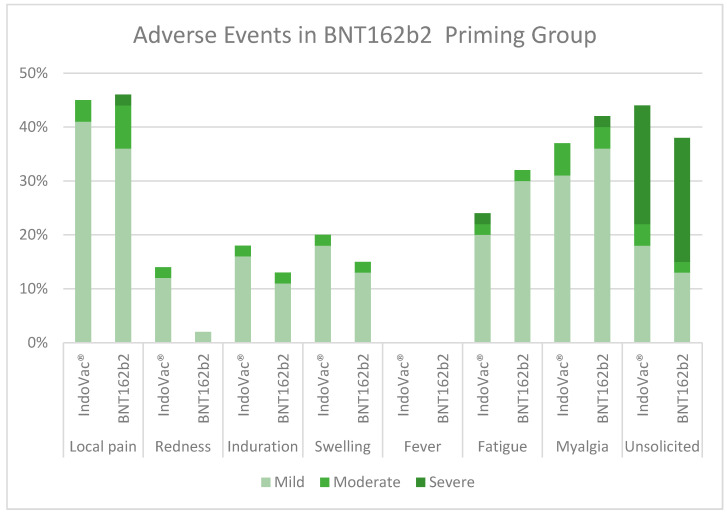
Adverse events in the BNT162b2 primary vaccine group.

## Data Availability

Data will be available on the main site of the study. Contact the author for future access.

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
