# Peer review of "Immunogenicity and Safety of SARS-CoV-2 Protein Subunit Recombinant Vaccine (IndoVac®) as a Booster Dose against COVID-19 in Indonesian Adults"

_vaccines, 2024, doi:10.3390/vaccines12050540_

Round 1
Reviewer 1 Report
Comments and Suggestions for Authors
This study is to evaluate benefits of the halal vaccine IndoVac® as a booster for COVID-19 vaccinations. This study is important for long-term immune protection.
1. In abstract, “this study planned for 900 individuals…” should be deleted, because in fact a total of 696 participants were enrolled. The 900 number, especially in abstract cannot provide any useful information, only cause misleading information or confusing.
2. The number of participants in primary mRNA vaccine (BNT162b2) group is only 93. As indicated by the authors, “the minimum sample size in the BNT162b2 primary vaccine group was not met because most of the population in the study area did not receive the BNT162b2 vaccine as their primary COVID-19 vaccine.” Can the results obtained in this group provide statistical useful information, or why the authors still want to include the data of this group in this report.
3. In the Introduction, the authors stated that “As a locally produced halal vaccine, IndoVac® has clear benefits for use in Indonesia [11,12,13].” How about other vaccines in comparing with IndoVac, regarding halal? More background information should be added to the introduction and the benefits should be discussed in the discussion section, if this is important.
4. All the 3 tables are not reviewable and should be reformatted.
5. In Discussion: Line 284-300: “Preclinical animal studies and controlled trials have demonstrated the potential utility of subunit protein vaccines, …. As RDB contains multiple conformational and conserved neutralising epitopes, the vaccine was predicted to have cross-neutralisation potency against variant viruses.” This is more like background information, maybe belong to Introduction.
6. The discussion section is not well organized and has some repeats. Editing is needed for this section and the entire manuscript.
Comments on the Quality of English LanguageEditing is needed.
Author Response
Thank you for your input, please see the attachement for the revisions

Reviewer 2 Report
Comments and Suggestions for Authors
Very interesting article on a different population with limited data in the literature. All graphs and tables are well selected and detailed. I would suggest improving the readability of the tables a little because the font is too small. The English is well written and the paragraphs structured in a way that grabs the reader's attention. I recommend publication after minor corrections concerning the tables so that they are more readable.
Author Response
Thank you for your input, please see the attachment for the revisions

Reviewer 3 Report
Comments and Suggestions for Authors
This is a comprehensive report on the boosting covid immune responses with a recombinant covid vaccine called IndoVac. It studied the boosting subjects previously vaccination with ChAdOx1, CoronaVac (Sinovac) and BNT162b2 except that the minimum sample size in the BNT162b2 primary group could not met because of the low number of subjects receiving this vaccine. Despite this and a slower response to boosting with BNT162b2 e the Indovac performed favourably with respect to immunogenicity and safety.
The conclusions are well supported by the results and the boosting had a very good long-term profile.
The assays used for antibody binding and neutralisation need to be specified in the paper.
Parts of the summary and introduction need to be checked for English because of the incorrect use of future tense.
There might need to editorial decisions made about the large amount of socio/economic data.
A good paper for a special issue and of considerable importance for a highly populated country
Comments on the Quality of English LanguageIssue with tense in abstract and introduction
Author Response

(The authors gave the same response as above.)

Round 2
Reviewer 1 Report
Comments and Suggestions for Authors
no further comments
Comments on the Quality of English Languageit is OK